# Smart Installation Weather Warning Decision Support

**Martin Tran [1], Samuel Kreinberg [1], Eric Specking [1], Gregory S. Parnell [1,*], Brenda Hernandez [1], Ed Pohl [1], George Gallarno [2], John Richards [2], Randy Buchanan [2] and Christina Rinaudo [2]**

[1] Department of Industrial Engineering, University of Arkansas, Fayetteville, AR 72701, USA; mltran@uark.edu (M.T.); stkreinb@uark.edu (S.K.); especki@uark.edu (E.S.); behernan@uark.edu (B.H.); epohl@uark.edu (E.P.)

[2] US Army Engineer Research and Development Center, Vicksburg, MS 39180, USA; george.e.gallarno@erdc.dren.mil (G.G.); john.p.richards@erdc.dren.mil (J.R.); randy.k.buchanan@erdc.dren.mil (R.B.); christina.h.rinaudo@erdc.dren.mil (C.R.)

* Correspondence: gparnell@uark.edu; Tel.: +1-914-720-3989

**Abstract:** Army installation commanders need timely weather information to make installation closure decisions before or during adverse weather events (e.g., hail, thunderstorms, snow, and floods). We worked with the military installation in Fort Carson, CO, and used their Weather Warning, Watch, and Advisory (WWA) criteria list to establish the foundation for our algorithm. We divided the Colorado Springs area into 2300 grids (2.5 square kilometers areas) and grouped the grids into ten microclimates, geographically and meteorologically unique regions, per pre-defined microclimate regions provided by the Fort Carson Air Force Staff Weather Officers (SWOs). Our algorithm classifies each weather event in the WWA list using the National Weather Service's and National Digital Forecast Database's data. Our algorithm assigns each event a criticality level: none, advisory, watch, or warning. The traffic network data highlight the importance of each road segment for travel to and from Fort Carson. The algorithm also uses traffic network data to assign weight to each grid, which enables the aggregation to the region and installation levels. We developed a weather dashboard in ArcGIS Pro to verify our algorithm and visualize the forecasted warnings for the grids and regions that are or may be affected by weather events.

**Keywords:** smart cities; smart installations; smart bases; weather modeling; decision analysis; influence diagrams



## 1. Introduction

The need to use integrated weather information (observations and forecasts from multiple sources) to provide warnings to reduce the risk to infrastructure and personnel has been described in the weather literature. Weather warnings require accurate and often real-time data that is available from anywhere. The works of Lakshmanan et al. created the second generation of the Warning Decision Support System—Integrated Information (WDSS-II) in the 2000s. WDSS-II brought forth advancements in weather detection, providing "greater temporal resolution and better spatial resolution" by operating on a network of radars rather than individual radars [1]. Since then, WDSS-II has been adopted by the National Severe Storms Laboratory, and which our usage of National Weather Service (NWS) and National Oceanic and Atmospheric Administration (NOAA) data may depend on. In recent years, the NWS has developed a novel approach for its warning systems. As storms initially form and progress throughout their life, they will take on many different shapes. Stumpf and Gerard envisioned a method to continuously update the storm's polygons to "mitigate gaps in warning coverage and improve the handling of storm motion changes" [2]. The Hurricane Forecast Improvement Program by the NOAA also paved the way in "improving tropical cyclone track and intensity predictions" by providing experimental models for tropical cyclone forecast [3]. Social media has also been leveraged

in providing weather conditions to both authorities and individuals. By using a heat map of social media posts pertaining to weather, researchers have proposed a model that estimates the situation and gives varying warning levels [4]. However, using social media also possesses its own issues. Colloquialisms and informal text styles make it difficult for researchers to accurately measure the situation. Shi et al. created several models that work together to extract the weather situation given in a body of text [5]. To communicate data interpretations effectively, applications often use Geographic Information Systems (GISs). GISs have long been utilized in many applications such as keeping the public safe from weather, to maintaining crops. Choi et al. created a decision support system (DSS) using GIS software (MapServer 2001) to manage watersheds and introduce a web-based GIS in the 2000s [6]. Short Message Service also provides a cheap alternative to communicating weather-related information [7].

Fort Carson, a United States Army Garrison, is in the Colorado Springs, CO, area. This geographic area frequently experiences several types of extreme weather. Table 1 provides extreme weather event information for within 50 miles of each of three Colorado locations (Fort Carson, Colorado Springs, U.S. Air Force Academy) from 1950 to 2010 [8]. Table 1 shows that hail, thunderstorm winds, floods, and heavy snow are the most frequent extreme weather events for Fort Carson and its surrounding area.

**Table 1.** Extreme Weather Event Data.

|  | Fort Carson | Colorado Springs | U.S. Air Force Academy |
|---|---|---|---|
| Hail | 1679 | 1806 | 1944 |
| Thunderstorm Winds | 242 | 261 | 235 |
| Flood | 139 | 158 | 152 |
| Heavy Snow | 41 | 41 | 43 |
| Winter Storm | 35 | 35 | 35 |
| Strong Wind | 14 | 14 | 13 |
| Wildfire | 10 | 10 | 10 |
| Blizzard | 7 | 7 | 7 |
| Winter Weather | 7 | 7 | 7 |

Severe weather negatively impacts employees and military operations. Identifying potential adverse weather conditions is critical for Fort Carson to avoid weather-related risks. For example, hail, thunderstorms, snow, and ice increase the travel risks for commuters and the movement of personnel and equipment during operations. The Fort Carson's Garrison Commander must make timely decisions to send employees home and close or delay non-mission essential base operations.

A review of the literature has supported the case that other researchers have performed similar research about weather classification and DSSs. Topics have varied from creating classification systems to supporting decision-making authorities. One such example is the works of Brandt et al. Their team created a weather rules-based decision aid that assists in military tactical operations routing. The application provides decision-makers with multiple paths to reach the end of the mission, highlighting grids with higher risks in yellow or red [9]. Aljohani et al. utilized machine learning to classify floods. With advances in the field of machine learning, the classification of specific weather hazards such as floods has reached a remarkably high accuracy in determining the danger levels in each area of the city [10]. This capability allows for quick responses to flood disasters to minimize human and monetary loss. New research has also improved existing flood forecasting by increasing accuracy and reducing error, especially for floods lasting longer than two hours [11]. And while machine learning and neural networks has advanced significantly, new research constantly updates the baseline. Chen et al. proposed a type of generative adversarial network that covers the vanishing gradient weakness of traditional deep learning methods [12]. Non-emergency DSSs also exist. Soil moisture impacts the Army's reliance on movement and logistics for its operations. To better support decision-making, the Army created their GeoWATCH

application to better understand the impacts of soil on operations [13]. Integrating DSS into other areas of research helps support individuals and businesses as well. Especially in agriculture, weather influences the yield of crops, and harnessing the weather forecasts influences decision-making on a day-to-day basis [14]. Singh et al. also created a similar system to help farmers in decision-making in India [15].

Decision support systems prove valuable in times of crisis. According to Na-Yemeh et al. [16], Oklahoma's OK-First weather DSS helped through generating $1.2 million in self-reported cost savings for 12 months. Research studies have developed user interfaces to support decision-makers with statistical summaries [17] and probabilistic forecasts [18] for rapid responses to weather conditions. Another style of DSS allows the user to reason about the impact of data, especially with the rising complexity in both the number and complexity of data sources [19]. In our research, we integrated a classification algorithm with a decision dashboard to support the garrison commander with weather information. The algorithm focuses on a rules-based classification method compared to using neural networks or machine learning. Since we already know the specific conditions of each of the weather criteria, we do not need to utilize neural networks and such. Our research differs from previous research by classifying weather events at varying resolutions: grid (2.5 square kilometers areas) level, region level, and installation level. With improved information, the commander can make decisions specific to each area without potentially disrupting other operations.

Our research has supported the Engineer Research and Development Center's (ERDC) Smart Installations research. The ERDC team has supported a public–private partnership with Fort Carson, U.S. Ignite, ERDC, West Point Operations Research Center, Mississippi State University, and the University of Arkansas. This smart installation project team had the overall goal to create a data-driven DSS for Fort Carson that provides improved information to support weather-related decision-making and communicate those decisions to all personnel. The paper is organized as follows. In Section 2, we describe the stakeholder analysis process and results. In Section 3, we identify the decisions required to accomplish our scope of work. Section 4 describes the weather data used in the project. In Section 5 we discuss the development of the decision support tool, including a comparison of different potential methods. Section 6 presents the mathematical models used in the logical weather classification algorithm. In Section 7, we describe the visualization software used to verify our algorithm, and Section 8 concludes with a summary and suggested future work.

## 2. Current Severe Weather Warning Process

To better understand the current process, we conducted extensive background research as well as a series of interviews. The team interviewed stakeholders involved in the Fort Carson adverse weather decision-making process to identify sources of information and understand the existing process.

The Weather Warning Company Commander was interviewed to understand the installation commander's needs and desired outputs. This interview helped us understand the current inclement weather call decision process, identify the most influential stakeholders, and identify the current data flow.

Figure 1 illustrates the existing labor-intensive decision process described in the interviews. This process involves several manual inputs and communications with operational and weather personnel and starts by identifying a possible inclement weather situation by Weather Officers (WOs). The Chief of Operations (CHOPS) gathers information from different divisions and meets with the principal divisions to review the potential adverse weather situation and the potential impact on operations. The Garrison Commander receives the weather assessments at meetings and determines the weather warning call decision. After deciding, they communicate their decision and the impact on installation operations.

The ERDC team then interviewed WOs who explained the technical part of the inclement weather call decision process. This process involved identifying possible adverse weather by collecting observable data from radars at Fort Carson Airfield, the U.S. Air

Force Academy, and the Colorado Springs Airport. In addition to using their local data sources, the WOs referenced several sources, such as information from the NWS and the NOAA. They then use a checklist, Fort Carson's Weather Warning, Watch, and Advisory (WWA) criteria list (Table 2), to analyze the observable data and forecasts to categorize potential severe weather events.

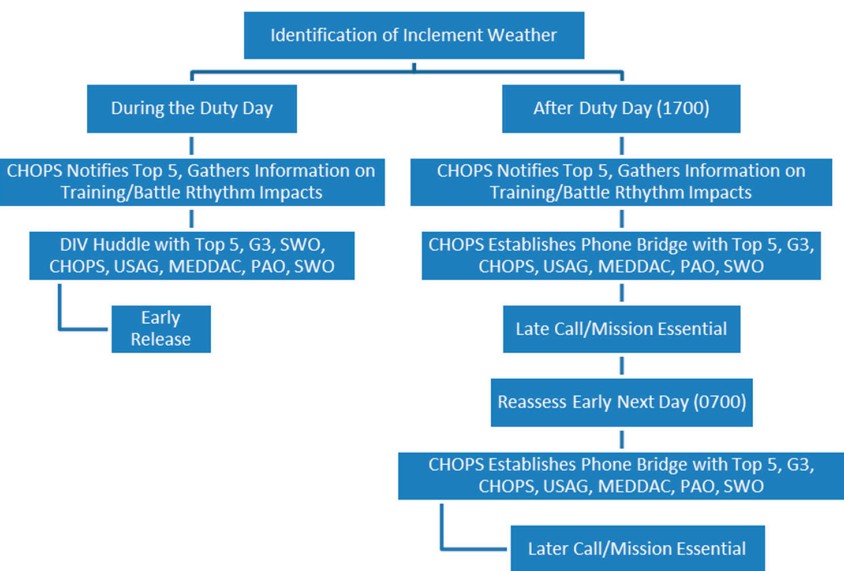

**Figure 1.** Inclement weather call process—process flow chart.

**Table 2.** Fort Carson WWA criteria.

| Severity Level | WWA List Details |
|---|---|
| Warning | Damaging Winds ≥ 45 knots are forecasted within the FC Reservation<br>Freezing Precipitation is forecasted for FC ReservationHeavy snow accumulation ≥ 4 inches within 12 h for FC Reservation<br>Observed lightning (within 5 NM of FC South)<br>Observed lightning (within 5 NM of FC North)<br>Observed lightning (within 5 NM of Butts Army Field)<br>Heavy rain ≥ 2 inches within 12 h for FC Reservation<br>Tornadoes expected (within 5 NM of FC North)<br>Tornadoes expected (within 5 NM of FC South)<br>Blowing Snow at FC Reservation. Visibility ≤ 1/4 square mile and wind ≥ 30 knots (34 mph) for ≥ 3 h<br>Moderate thunderstorms for the FC reservation (high wind ≥ 35 knots (40 mph) but <45 knots (52 mph) and/or large hail ≥ 1/4 inch but <1/2 inch)<br>Severe thunderstorms for the FC Reservation (damaging hail ≥ 1/2 inch and/or damaging wind ≥ 45 knots) |
| Watch | Potential for damaging surface winds ≥ 45 knots exists for the FC Reservation<br>Potential for freezing precipitation exists for the Fort Carson Reservation<br>Potential heavy rain ≥ 2 inches within 12 h exists for the Fort Carson Reservation<br>Potential for heavy snow accumulation ≥ 4 inches within 12 h exist for FC Reservation<br>Potential for lightning for FC North within 5 NM<br>Potential for lightning for FC South within 5 NM<br>Potential for lightning within 5 NM of Butts Army Airfield<br>Potential tornado within FC South<br>Potential for blowing snow at FC Reservation (visibility ≤ 1/4 sq mi and winds ≥ 30 knots (34 mph) for ≥3 h<br>Potential for moderate thunderstorms for FC Reservation (high wind ≥ 35 (40 mph) but <45 knots (52 mph) and/or large hail ≥ 1/4 inch but <1/2 inch)<br>Severe thunderstorms for the FC Reservation (potential for damaging hail ≥ 1/2 inch and/or damaging wind ≥ 45 knots (52 mph))<br>Potential for tornado exists (tornado within FC North) |
| Advisory | Forecasted strong winds > 34 knots (39 mph) but <45 knots (52 mph) for FC Reservation<br>Observed winds ≥ 24 knots but <35 knots on FC Reservation |

The WOs split the WWA list into three categories: warning, watch, and advisory. In terms of severity, warning and watch represent the same severity level. Advisory conditions represent milder conditions. We also define these three categories by the time that they occur as we receive our weather data. Upon receiving the data, we define weather conditions as warnings when they will occur within one hour (known as the "nowcast") and watches when they will occur any time past the next hour (the forecast). Advisory conditions have "nowcast" and forecast conditions.

## 3. Project Scope

We needed to identify the scope of our work since the project consisted of several geographically separated teams working together to create the DSS. We used a decision hierarchy [20] to identify and categorize the decisions our team had to make. A decision hierarchy divides the project decisions into three categories: (1) the decisions that the decision-makers, stakeholders, and team have made, (2) the decisions that our team must make, and (3) the subsequent decisions that will be made by decision-makers, stakeholders, and the team.

This decision hierarchy helped us understand our algorithm development decisions. In the first months of the project, we had weekly project meetings and special meetings with U.S. Ignite, Fort Carson, and the ERDC team members. We developed the decision hierarchy based on these meetings and updated as the project progressed. The decision hierarchy, illustrated in Figure 2, provides the decisions that were made (top of the figure), the major team decisions (middle of the figure), and the subsequent decisions (bottom of the figure).

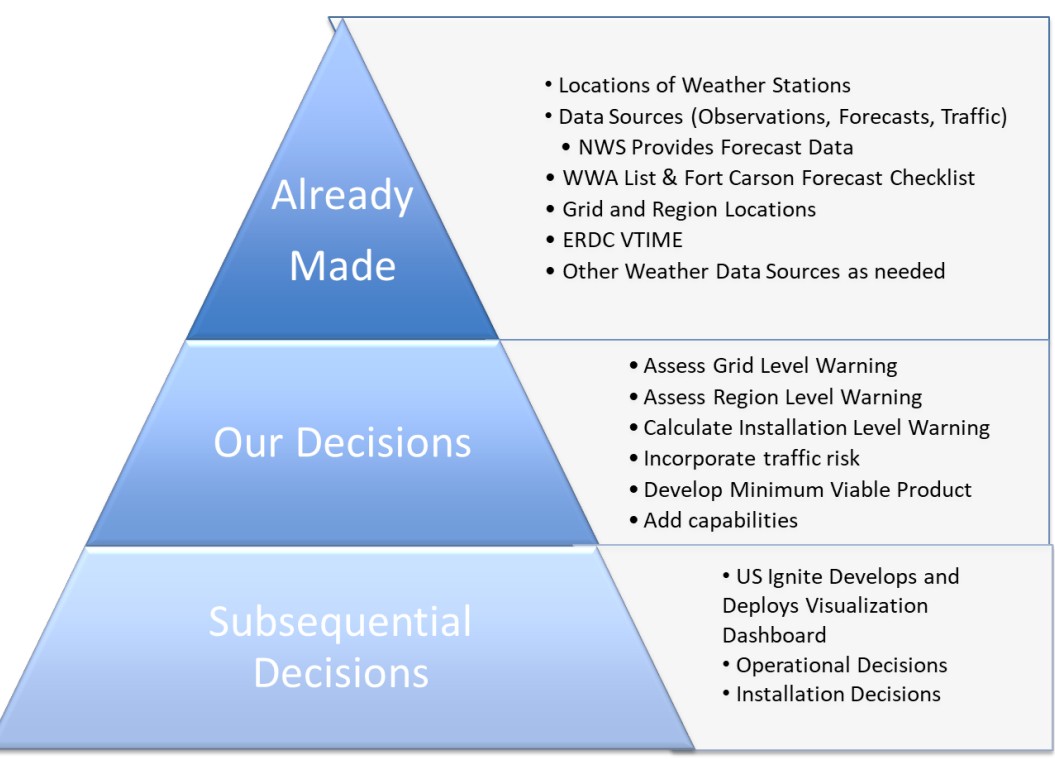

**Figure 2.** Fort Carson weather warning decision hierarchy.

We made several key decisions early in the project. First, we selected the NWS as the project's primary data source. The NWS uses a suite of weather models ranging from local, high-resolution, short-term models (the High-Resolution Rapid Refresh model (HRRR)), medium resolution models that produce quality short-range forecasts for North America (North American Model (NAM)), to their Global Forecast System (GFS) [21]. Second, we would use the Fort Carson WWA Criteria List to model the weather warning process currently used at Fort Carson and other installations. Third, we would evaluate the criteria

at the grid level (i.e., 2.5 by 2.5 km squares provided by NWS), microclimate, and base levels. The ERDC divided the Colorado Springs region into 10 microclimates with different weather conditions due to different geography and environments. These microclimates align with WOs' required reporting areas. Finally, the ERDC plans to migrate the new tool to Virtual Tool for Installation Mission Effectiveness (VTIME) capabilities to enable other Army installations to use the applications.

Our major decision objectives were to identify the weather observations and prediction data needed for weather warnings and to develop and verify a decision support algorithm to determine the potential weather warning at the grid, region, and base levels. The subsequent decisions include the development of the U.S. Ignite visualization dashboard, the future operational decisions, and the future installation weather warning decisions.

## 4. Weather Data

After understanding our scope, we needed to understand the various weather and weather data sources.

### 4.1. Data Source Identification

Since installation stakeholders heavily rely on publicly accessible weather data and road condition estimates, the ERDC investigated weather data sources that we could use in addition to the already available government weather data. One of the primary non-Department of Defense data sources was NOAA as their data feed into other sources, such as the following:

- NWS Forecast API;
- NOAA NDFD Forecast;
- NOAA High-Resolution Rapid Refresh (HRRR) model;
- Real-time NWS Products;
- Local Climatological Data (LCD);
- On-site weather stations.

After identifying weather data sources to meet the needs of the WWA criteria list, we decided to use data from the NWS Forecast API. The NWS Forecast API provides forecast data for a 2.5 km grid square and is the baseline for synchronizing subsequently collected weather data. We exported and created a sample forecast dataset, to provide the data needed by the Fort Carson weather team. These data contained 57 weather attributes, such as precipitation measures, temperature measures, and hazard determinants. The dataset contained these weather attributes for each grid for the current time to 5 days out in hourly increments, comprising ten regions and 2299 grids. Each grid is 2.5 km by 2.5 km with some grids being in multiple regions.

### 4.2. Weather Station Locations

We selected the NWS Forecast API for use, based on data availability and cost. These data use the observation values produced by weather stations. Therefore, we found the locations of the weather stations used by the NWS to forecast the weather for each grid in our 10 regions. We used Python to read the data that contains the coordinates of the grids and used the midpoint of each of the grids to create URLs for the API to identify the weather stations. The API generated links for each weather station that subsequently provided data for the region.

We found that 53 unique weather stations provided data to predict the weather for the 2299 grids, shown in Table 3. Additionally, we found that six unique weather stations were physically located in one of the 10 regions. This means we are only able to obtain observed data for six grids within the region.

We plotted the unique weather stations to gain additional insights, shown in Figure 3. The 22 purple points represent weather stations that provide data for the Monument and North, Northeast, Woodland Park, and Colorado Springs East /Airport regions. The 31 gold points provide additional data for the weather stations that are in all 10 regions.

From this visualization, we gained a better understanding of the physical locations of the weather stations that provide the weather forecast data for the Colorado Springs area.

**Table 3.** Weather stations used in data.

| Region | Weather Stations Providing Data | Weather Stations within Region |
|---|---|---|
| Colorado Springs | 31 | 0 |
| Colorado Springs East/Airport | 53 | 2 |
| Fort Carson | 31 | 1 |
| Fountain | 31 | 0 |
| Monument and North | 53 | 0 |
| Northeast | 53 | 1 |
| Pinion Canyon | 31 | 0 |
| Pueblo | 31 | 1 |
| USAFA | 31 | 1 |
| Woodland Park | 53 | 0 |

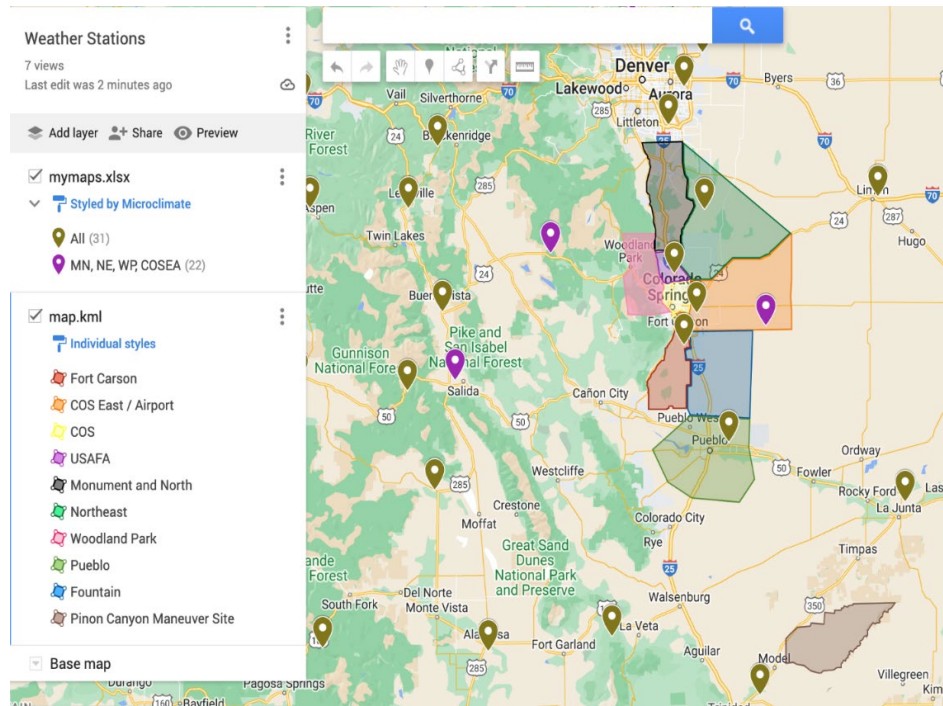

**Figure 3.** Region layout and weather station locations.

*4.3. National Weather Service Forecast Data*

We mapped the NWS Forecast API data to the WWA criteria list to determine what data could help classify weather types. This investigation revealed 16 of the 27 WWA criteria align with the NWS data. Table 4 contains the 11 variables from the NWS Forecast API that enable the classification of the 16 WWA criteria. We performed unit conversions to put the WWA criteria and weather data in the same units. For example, the NWS Forecast API provides wind speed in kilometers per hour, while the WWA criteria has wind speed in knots.

We were unable to classify the remaining 12 WWA criteria, such as hail or tornados, because the NWS Forecast API measures surface to low atmospheric conditions. For example, hail prediction requires atmospheric weather data since it impacts the size of hail formation [22].

We investigated severe weather to find other ways to classify the rest of the WWA criteria. We found that indicators such as the Convective Available Potential Energy (CAPE) value are key to predicting thunderstorms, hail, and tornadoes. However, during our research, we found that predicting tornadoes proves to be the most difficult to determine. As a result, the complexities with the tornado prediction led us to look for another data

source that could provide us with what we need to classify the remaining 12 WWA criteria. We further explore our tornado research in our technical report [23].

**Table 4.** List of NWS-provided weather attributes for WWA criteria identification.

| Weather Attribute | Units |
|---|---|
| Temperature | Celsius |
| Probability of X wind speed | Percent |
| Wind speed | Km/h |
| Probability of X wind gust speed | Percent |
| Wind gust speed | Km/h |
| Ice accumulation | Millimeters |
| Quantitative precipitation | Millimeters |
| Probability of precipitation | Percent |
| Snowfall amount | Millimeters |
| Lightning activity level | Scale for lightning activity (1–6) |
| Probability of thunderstorms | Percent |

### 4.4. NOAA NDFD Data

We decided to use the NDFD data as a secondary dataset and contains the convective weather outlook variables, which are useful in identifying severe weather events such as hail and tornadoes. The NDFD data refreshes every half hour [24], which allows it to be particularly valuable in early warnings. However, we chose not to implement some of the NDFD variables because the NDFD listed many of them as experimental and historically do not populate at every data pull. Therefore, we implemented the non-experimental values associated with Convective Weather Outlook of the NDFD. The Convective Weather Outlook variable produces a number from 0 to 148 to signify a hazard [25]. Out of the 148 available codes, the team found 6 [see Table 5] aligned with the Fort Carson WWA criteria we needed. These codes enabled us to classify eight additional WWA criteria.

**Table 5.** NDFD Convective Hazard Outlook.

| Code Number | Label | Documentation |
|---|---|---|
| 0 | None | |
| 2 | Tornado Warning | A Tornado Warning is issued when a tornado is imminent. When a tornado warning is issued, seek safe shelter immediately |
| 24 | Blizzard Warning | A Blizzard Warning means that the following conditions are occurring or expected within the next 12 to 18 h. (1) Snow and/or blowing snow reducing visibility to ¼ mile or less for 3 h or longer. AND (2) Sustained winds of 35 mph or greater or frequent gusts of 35 mph or greater. There is no temperature requirement that must be met to achieve blizzard conditions. |
| 46, 47 | Tornado Watch | A Tornado Watch is issued when severe thunderstorms and tornadoes are possible in and near the watch area. It does not mean that they will occur. It only means they are possible. Severe thunderstorms are defined as follows: (1) Winds of 58 mph or higher (2) AND/OR Hail 1 inch in diameter or larger. |
| 113 | Blizzard Watch | A Blizzard Watch is issued when conditions are forecasted to last 3 h or longer. (1) Sustained winds greater than 35 mph (2) Falling or blowing snow reducing visibility to ¼ mile or less |

Although we have been able to identify data to determine the presence of tornados and blizzards using the NDFD dataset, hail remains unclassifiable since we do not have CAPE values. Without CAPE values or other means to calculate hail size, we are currently unable to classify four of the WWA criteria.

### 4.5. Combining NWS and NDFD Data

With the data identified, we created a system to automatically retrieve the NWS Forecast and NDFD Convective Hazard Outlook for storage in a database. The NWS Forecast uses an application interface to retrieve the data in hourly time frames for the desired locations. The NDFD data comes in GRIB2 files that must be downloaded. This meant that the team had to schedule hourly downloads of the data and process it to connect it to the NWS data based on the grid, day, and time. We combined the datasets and stored them in a local database. Then, we pull the data and store it in a data frame, which is a two-dimensional tabular data structure like an Excel spreadsheet [26], for our classification algorithm to use. We visualize this process in Figure 4.

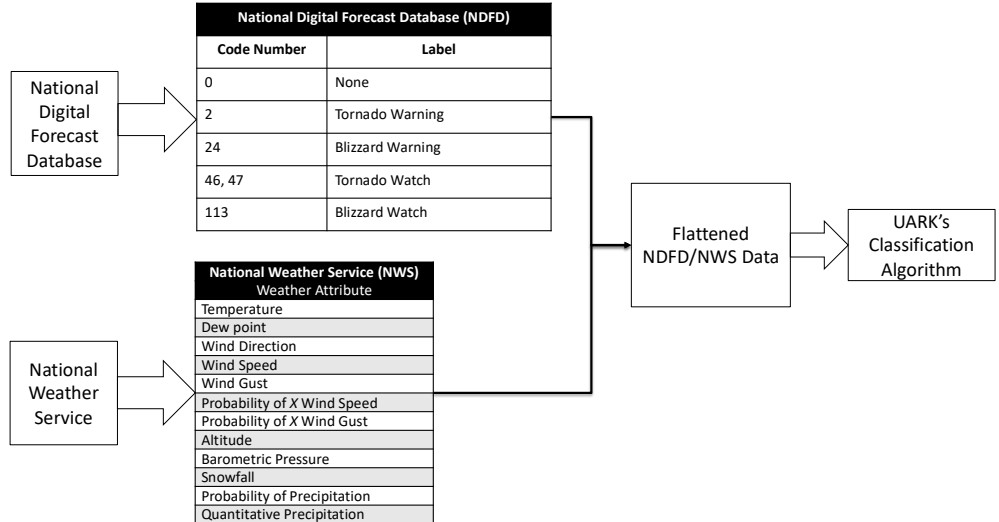

**Figure 4.** NWS and NDFD data aggregation method overview.

## 5. Weather Decision Support Tool

### 5.1. Interpreting the Modeling Process

With a better understanding of our scope and the data, we created an influence diagram (ID), illustrated in Figure 5, to demonstrate our understanding of the problem. IDs are a decision analysis tool that identifies certain and uncertain information, the decisions and the values associated with a decision process, and provides the flow of information and the probabilistic relationships among these variables [20]. The circles represent the variables that are uncertain. The IDs represent the decisions as squares and octagons as the value nodes. This ID allowed our integrated framework to model our decision problem. We analyzed the weather forecast data and observation data grid level and aggregated it to the region (microclimate) and base levels for installation decision-making. Making the decision was not our aim. Rather, we provided information to help the decision-maker make a data-informed decision based on the weather conditions, their current operations, and their planned operations.

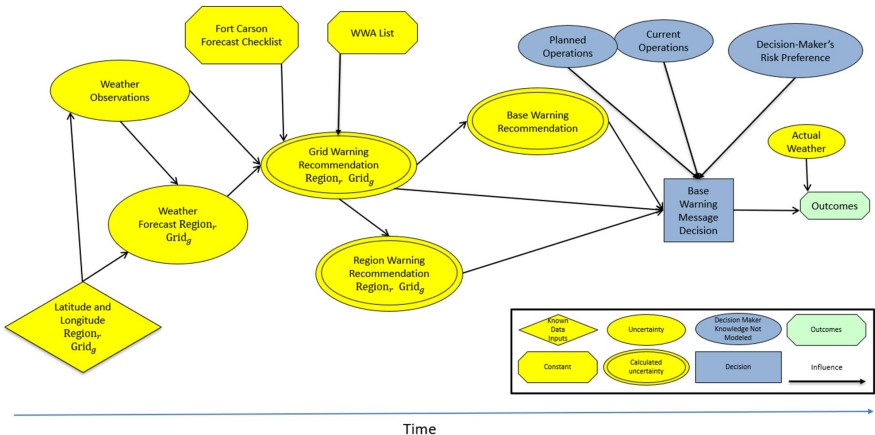

**Figure 5.** Fort Carson weather warning decision influence diagram.

## 5.2. Weather Classification Algorithm

The logical classification algorithm, illustrated in Figure 6, has 10 functions, and uses the indices and constants stated in Table 6. We started by reading the flattened data and a grid importance input file and sequentially called our first function to process grid level data to produce a grid level output. This function then called two sub-functions. The first sub-function, process_wwa(), examined the weather data for each time and grid combination to compare its weather data to the WWA criteria list. It set presence$_{t,g,c}$ based on Equation (1). For example, if the precipitation quantity was greater than 2 inches, then the model set the flag to a 1 for the output flag that corresponds to a weather watch for heavy rain. The second sub-function, process_events(), stored the output of the previous function to help with region- and installation-level aggregations.

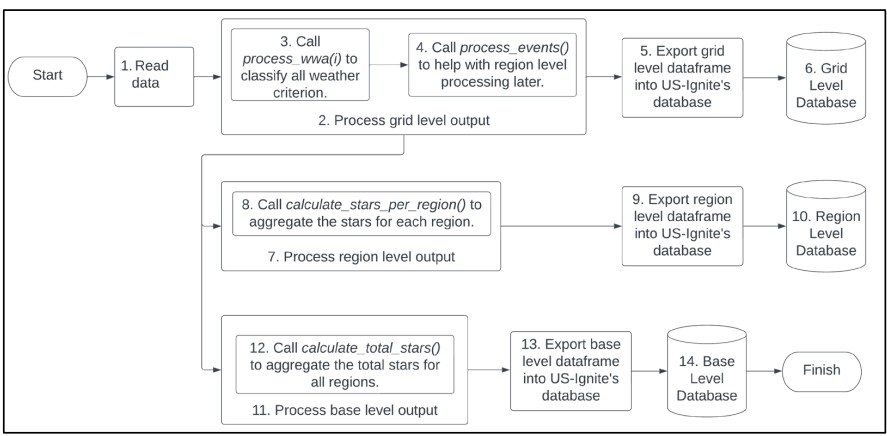

**Figure 6.** Classification algorithm flow chart.

**Table 6.** Classification algorithm indices and constants.

| Variable Type | Variable | Description |
|---|---|---|
| Index | g | Grid index = {1, 2, . . ., 2299} |
| Index | r | Region = {1, 2, . . ., 10} |
| Index | t | Time in hours |
| Index | c | WWA criteria = {1, 2, . . ., 22} |
| Constant | G | Total number of grids = 2299 |
| Constant | $N_r$ | number of grids in region r in the Colorado Springs area |
| Constant | $B_c$ | number of grids for the base criteria c |
| Constant | $grid\_score_g$ | Grid importance score = {1, 2, 3, 4, 5} |

$$presence_{t,g,c} = \{0; 1\}: 1 \text{ if criteria c is satisfied at time t in grid g or else } 0 \qquad (1)$$

Before we saved the grid level data, we used the grid importance input file to create grid importance weights. We created this input file by assigning a grid importance score from 1 to 5 based on the criteria shown in Table 7 as well as past traffic flow, population, and geographic data to assign scores, which we illustrated in Figure 7. We normalized these grid scores into the grid's importance weight by using Equation (2) and merged it with the grid level weather data. We then called the region and installation level functions, which we independently called or called in parallel. The most important thing was that the grid level functions were complete before the region and installation functions were called.

$$grid\_weight_g = grid\_score_g / \sum_1^G grid\_score_g \qquad (2)$$

**Table 7.** Grid importance score criteria.

| Score | Description | Color |
|---|---|---|
| Most Critical 5 | geographical grids on the Fort Carson installation | Black |
| 4 | geographical grids on major highways or areas near Fort Carson | Red |
| 3 | geographical grids on highways that feed major highways or areas close to the Fort Carson installation | Purple |
| 2 | geographical grids in residential areas or areas on highways that are not that near to the Fort Carson | Green |
| 1 Least Critical | all other geographical grids | White |

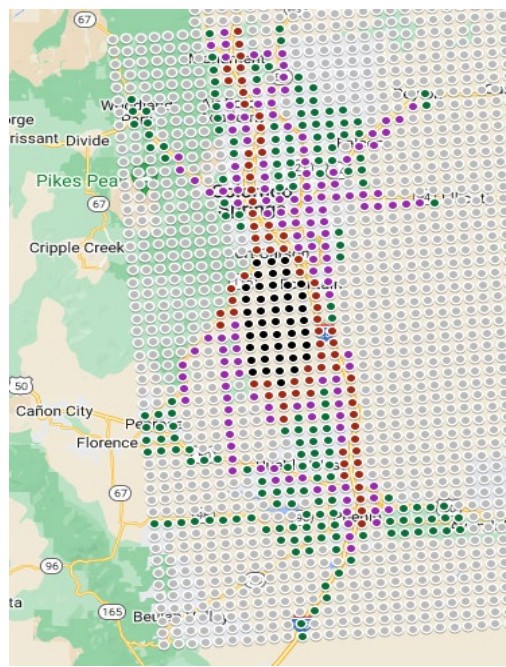

**Figure 7.** Grid importance in Colorado Springs.

The function to process the region level output provided a weighted region score using the weighted grids impacted by each criteria c and divided it by the total grid weight for that region, shown in Equation (3). If the forecast for grid g at time t did not meet the criteria c for any grid in the region, the region score was zero. If the forecast met the criteria

c for every grid in the region, the region score was 1. This function produced a region score, or a weighted sum, which showed how much some criteria impact some regions.

$$region\_score_{t,r,c} = \sum_{g=1}^{N_r} \frac{presence_{t,g,c} \times grid\_weight_g}{\sum_{g=1}^{N_r} grid\_weight_g} \tag{3}$$

We created the installation level output similarly to the region level method by calculating the total percentage of grids impacted by each weather criteria by all the grids, shown in Equation (3). On an important note, this warning probability was the percentage of grids affected by each weather criterion.

$$base\_score_{t,c} = \sum_{g=1}^{B_c} \frac{presence_{t,g,c} \times grid\_weight_g}{\sum_{g=1}^{B_c} grid\_weight_i} \tag{4}$$

After we created the outputs for each level (i.e., grid, region, and installation), we sent the data to US-Ignite who stored it in a database for their system to create a visualization dashboard and produce notifications.

We had to make several assumptions when developing the classification algorithm. First, we assumed that all criteria could occur at the same time. Obviously, this was close to impossible to happen, but more importantly, it allowed us to classify multiple conditions at once, such as lightning and heavy rain. Next, the WWA criteria list has lightning criteria for within a 5 nautical mile (NM) radius of a targeted location. We simplified this by using a square radius instead of the traditional circle. This simplified our code since our weather sources provide data in square grids, which aligned with the longitudinal or latitudinal values. Finally, our data sources contained the probabilities of some conditions, such as the probability of wind gusts over 50 miles per hour. We classified any probability above zero. Future research should look at the impact of adjusting these assumptions. We implemented our code using Python and several Python packages such as pandas and NumPy for data manipulation and datetime, pytz, and math for date and time calculations.

## 6. Results

Verification is an important part of any system design. Therefore, we developed a method to verify that the classification algorithm developed does what we logically expected. To accomplish this, we started by creating a notional dataset that assigned each of the WWA criteria to an associated time period with data to flag the associated time WWA criteria for all grids. We then processed the verification test file with our classification algorithm, which produced three output files. We assumed that any weighted ratios over zero will color that region or the installation the correct color. For example, if watch_heavyRain has a weighted ratio of 0.01, then the ArcGIS code will still color it red. After producing the output files, we then created ArcGIS files to process each output file to independently verify the classification algorithm was triggering the conditions correctly. We did this by assigning the output to a BRAG (i.e., black, red, amber, or green) color based on Table 8.

**Table 8.** BRAG classification.

| Color | WWA Severity |
| --- | --- |
| Black | Warning |
| Red | Watch |
| Amber | Advisory |
| Green | None |

This process proved to be helpful. It helped us find a few coding issues in our classification algorithm. For example, we found that the algorithm reversed the lightning watch criteria for Fort Carson North and Fort Carson South. We had flipped the hard-coded

longitude and latitude boundaries. In other words, the northern points were flipped with the southern points. This process verified all 22 criteria for the grid, region, and installation levels as shown in Figure 8 below. Tran et al. [23] documents the verifications for each of the 22 criteria for each level. For example, at 8:00 pm on 12 August, we expect that the watch_surfaceWindsO45 condition would trigger. In ArcGIS at the time and date, the condition triggered, coloring the grids/regions/installation in red as shown in Figure 9.

| | WWA List Details | Data Requirement/Coding | Verification Status |
|---|---|---|---|
| **Warning** | Damaging Winds >=45 knots are forecasted within the FC Reservation | COMPLETE | VERIFIED |
| | Freezing Precipitation is forecasted for FC Reservation | COMPLETE (6/26) | VERIFIED |
| | Heavy snow accumulation >= 4 inches within 12 hours for FC Reservation | COMPLETE | VERIFIED |
| | Observed lightning (within 5 NM of FC South) | COMPLETE | VERIFIED |
| | Observed lightning (within 5 NM of FC North) | COMPLETE | VERIFIED |
| | Observed lightning (within 5 NM of Butts Army Field) | COMPLETE | VERIFIED |
| | Heavy rain >= 2 inches within 12 hours for FC Reservation | COMPLETE | VERIFIED |
| | Tornadoes expected (within 5 NM of FC North) | COMPLETE (8/14) | VERIFIED |
| | Tornadoes expected (within 5 NM of FC South) | COMPLETE (8/14) | VERIFIED |
| | Blowing Snow at FC Reservation. Visibility <= 1/4 square mile and wind >= 30 knots (34 mph) for >= 3 hours | COMPLETE (8/14) | VERIFIED |
| | Moderate thunderstorms for the FC reservation (high wind >= 35 knots (40 mph) but < 45 knots (52mph) and/or large hail >= 1/4 inch but < 1/2 inch) | NEED CAPE VALUE FROM GFS | CAN'T START |
| | Severe thunderstorms for the FC Reservation (damaging hail >= 1/2 inch and/or damaging wind >= 45 knots) | NEED CAPE VALUE FROM GFS | CAN'T START |
| **Watch** | Potential for damaging surface winds >= 45 knots exists for the FC Reservation | COMPLETE | VERIFIED |
| | Potential for freezing precipitation exists for the Fort Carson Reservation | COMPLETE (6/26) | VERIFIED |
| | Potential heavy rain >= 2 inches within 12 hours exists for the Fort Carson Reservation | COMPLETE | VERIFIED |
| | Potential for heavy snow accumulation >= 4 inches within 12 hours exist for FC Reservation | COMPLETE | VERIFIED |
| | Potential for lightning for FC North within 5 NM | COMPLETE | VERIFIED |
| | Potential for lightning for FC South within 5 NM | COMPLETE | VERIFIED |
| | Potential for lightning within 5 NM of Butts Army Airfield | COMPLETE | VERIFIED |
| | Potential tornado within FC South | COMPLETE (8/14) | VERIFIED |
| | Potential for blowing snow at FC Reservation (visibility <= 1/4 sq mi and winds >= 30 knots (34 mph)for >= 3 hours | COMPLETE (8/14) | VERIFIED |
| | Potential for moderate thunderstorms for FC Reservation (high wind >= 35 (40 mph) but < 45 knots (52 mph) and/or large hail >= 1/4 inch but < 1/2 inch) | NEED CAPE VALUE FROM GFS | CAN'T START |
| | Severe thunderstorms for the FC Reservation (potential for damaging hail >= 1/2 inch and/or damaging wind >= 45 knots (52 mph)) | NEED CAPE VALUE FROM GFS | CAN'T START |
| | Potential for tornado exists (tornado within FC North) | COMPLETE (8/14) | VERIFIED |
| **Advisory** | Forecasted strong winds > 34 knots (39 mph) but < 45 knots (52 mph) for FC Reservation | COMPLETE | VERIFIED |
| | Observed winds >= 24 knots but < 35 knots on FC Reservation | COMPLETE | VERIFIED |

**Figure 8.** Identifiable criteria for the Fort Carson WWA list.

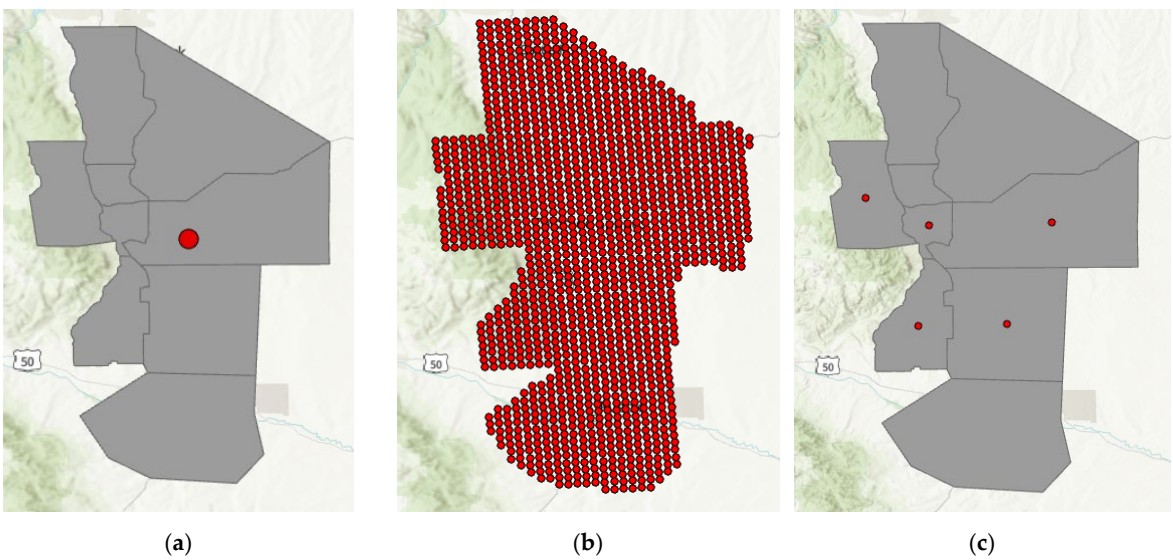

(**a**)    (**b**)    (**c**)

**Figure 9.** Triggering the grid, region, and installation levels for watch_surfaceWindsO45 condition. (**a**) Installation level; (**b**) grid level; (**c**) region level.

## 7. Summary

The United States Army Smart Installation initiative utilizes weather information and technology to help make informed decisions such as closing the installation due to adverse weather conditions. The outcomes of this research support the Garrison Commander in sustaining efficient installation operations with a small impact to the risk and safety of its personnel. While classifying weather events is critical to this research, the level of detail calculated at the grid, region, and installation levels helps the installation to potentially make better decisions, such as just dismissing only certain regions as opposed to all regions of interest. In addition to the different resolution levels, we captured risk preference with grid importance scores, which we normalized before using in the classification algorithm. By scoring each grid, we can evaluate the criticality of any weather event occurring with respect to how it may affect installation operations. We verified that the classification algorithm correctly classified the 22 implemented WWA criteria at the grid, region, and installation level.

## 8. Research Contributions

While our research is not completely novel in the field of weather decision support systems, we have accomplished four major goals towards developing a decision aid for the Army installation commander. First, we developed a data pipeline with the NDFD and NWS data that is preprocessed into a single flattened file. Second, we developed a rules-based classification algorithm that calculates the presence of a set of weather events with specified grids and regions of interest. The algorithm also determines the ratio of the base affected by each of the weather events. To verify that our algorithm correctly classifies the weather events, we used ArcGIS and provided visualizations of our testing as part of the verification process. Third, our partnership with US-Ignite will use our algorithm to develop the visualization dashboard. Fourth, this concept will be utilized at other Army installations using the Virtual Testbed Installation for Mission Effectiveness (VTIME), the Army's cloud-based application.

## 9. Limitations and Challenges

Some of the limitations and challenges include not being able to classify four of the WWA criteria, not implementing any of the observation data from the Army's ground weather sensors, and that other installations may have a different WWA list to that of Ft. Carson's. The four WWA criteria require knowledge of hail size; however, without the CAPE value, we cannot calculate the hail size for the thunderstorm criteria. We could not implement the observation data from the ground weather sensors due to timing. Finally, our research focuses on Ft. Carson only. While other installations can adopt Ft. Carson's intersecting WWA criteria, they will have to develop their own set of rules for the classification of new criteria.

## 10. Future Work

Future work could include adding CAPE to the flattened dataset to enable the classification of all the Fort Carson weather warning criteria. To reduce errors in cases such as false positives, we could incorporate real-time observations into the classification algorithm. Our algorithm currently uses a "nowcast" to classify weather for the current time-period, but the ERDC and U.S. Ignite have access to on-installation weather station data for true real-time observational data. One way to use these data is to compare the weather station data to the grid "nowcast" and forecast data to include data accuracy. Another opportunity area would be to look at how our assumptions impact the decision. Finally, other military installations could be considered for demonstrating the feasibility of transitioning and adapting the presented classification algorithm.

**Author Contributions:** Conceptualization, R.B. and G.S.P.; methodology, E.S., M.T. and G.S.P.; software, M.T.; validation, M.T. and S.K.; formal analysis, G.S.P. and E.S.; investigation, M.T., B.H.,

G.S.P. and E.S.; resources, G.S.P.; data curation, M.T. and G.G.; writing—original draft preparation, M.T.; writing—review and editing, M.T., E.S., G.S.P. and R.B.; visualization, S.K. and M.T.; supervision, E.S., G.S.P., J.R. and R.B.; project administration, E.S., G.S.P., E.P., J.R., C.R. and R.B.; funding acquisition, R.B., C.R. and G.S.P. All authors have read and agreed to the published version of the manuscript.

**Funding:** The U.S. Army Engineer Research and Development Center (ERDC) funded this research, under grant number W81EWF22008495.

**Data Availability Statement:** Figure 4 provides the data sources and integration of the dynamic data. Data are available from these government sources: the National Weather Service (https://www.weather.gov/documentation/services-web-api (accessed on 9 September 2022)) and the National Digital Forecast Database (https://www.ncei.noaa.gov/products/weather-climate-models/national-digital-forecast-database (accessed on 1 December 2022).

**Acknowledgments:** We would like to acknowledge the help of the ERDC team for the contributions to the data curation and processing steps and project management.

**Conflicts of Interest:** The authors declare no conflict of interest.

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
