# Peer review of "Smart Installation Weather Warning Decision Support"

_systems, doi:10.3390/systems12010014_

Round 1

Reviewer 1 Report

Comments and Suggestions for Authors

This paper proposes a decision support system to identify several types of extreme weather on Colorado. The results seem to be promising.

The paper can be improved taking into account the following aspects:

The bibliographic references in the state of the art are very limited and must be expanded.

Figure 1 is challenging to comprehend as several acronyms have not been explained in the paper.

Table 5 does not describe the 'blizzard watch' label.

Author Response

This paper proposes a decision support system to identify several types of extreme weather on Colorado. The results seem to be promising.

The paper can be improved taking into account the following aspects:

Thank you for your review.  We have addressed each aspect below.

1. The bibliographic references in the state of the art are very limited and must be expanded.

  • Need more journal-type references
    • Google Scholar
    • MDPI
    • Web of Science

Response: We now begin our paper with a summary of the literature that identifies the need to use integrated weather information (observations and forecasts from multiple sources) to provide warnings to reduce the risk to infrastructure and personnel. We’ve also added articles about different approaches to weather warning decision making.

2. Figure 1 is challenging to comprehend as several acronyms have not been explained in the paper.

Response: Every acronym is used and described in the paper.

3. Table 5 does not describe the 'blizzard watch' label.

Response: Blizzard watch is now described in the paper.

Reviewer 2 Report

Comments and Suggestions for Authors

This manuscript brings a classification algorithm integrated into a decision dashboard presenting weather information using the Weather Warning, Watch, and Advisory criteria list and analyzing micro-climate regions provided by Fort Carson (USA) Weather Officers selecting, among multiple sources, data from National Oceanic and Atmospheric Administration (NOAA) National Digital Forecast Database (NDFD) and combining them with National Weather Service (NWS) data.

The authors presented the workflow involved in the decision-making process according to information collected using interviews with weather personnel, to support understanding of the involved workflow, and the project the structure in three layers.

The classification algorithm developed is presented in the form of a flow chart, using flattened data and a grid importance input file, processing these data into a grid-level output. The results are presented in a GIS dashboard developed in ArcGIS software.

The study developed is quite interesting and has good applicability in practice to support weather conditions forecasting, not only for military proposals but expanding it for any kind of applications, for instance, in the agriculture area.

Despite it represents an important technological architecture for weather studies and applications, the authors can consider my recommendations below to support improving their text:

1.       The authors should expand their theoretical references. Currently, there are only 13 works cited, and half of them are from recent years (2020 to 2023). Please, expand and update your references.

2.       Did you compare your algorithmic approach with other previous existing ones? I recommend using the evidence in the literature to support it. If you produce performance metrics by testing your algorithm, you can provide it here to compare to other classification algorithms presented in the literature.

3.       Please, give some details of your technological platforms, especially involving other software than ArcGIS. Which technology did you use to implement your algorithm? Which programming language, for instance?

4.       Despite I understood the core contribution of the article is the classification algorithm, I missed the presentation of more related results. The results part is only focused on basic presentations of the grid in a GIS dashboard. If you have any numerical results from the algorithm application that can be presented in your text, and any other graphs, please provide them in your text with the due explanations.

5.       What are the main research contributions? You can divide them into theoretical and practical implications, for instance. Try to evidence:

a.       What does your research improve regarding the existing knowledge in classification algorithms to climatical conditions?

b.       What is the most innovative element (or elements) in your proposal, that differs from what already exists in already existing similar frameworks?

c.       What are the main contributions of your proposal for military applications?

d.       How can you apply your proposal to other areas than the military?

6.       Describe the current limitations, challenges, and difficulties in the process involved in implementing and applying your framework. You can insert this description in a Final Remarks section, also incorporating the Future Work section in it.

Author Response

This manuscript brings a classification algorithm integrated into a decision dashboard presenting weather information using the Weather Warning, Watch, and Advisory criteria list and analyzing micro-climate regions provided by Fort Carson (USA) Weather Officers selecting, among multiple sources, data from National Oceanic and Atmospheric Administration (NOAA) National Digital Forecast Database (NDFD) and combining them with National Weather Service (NWS) data.

The authors presented the workflow involved in the decision-making process according to information collected using interviews with weather personnel, to support understanding of the involved workflow, and the project the structure in three layers.

The classification algorithm developed is presented in the form of a flow chart, using flattened data and a grid importance input file, processing these data into a grid-level output. The results are presented in a GIS dashboard developed in ArcGIS software.

The study developed is quite interesting and has good applicability in practice to support weather conditions forecasting, not only for military proposals but expanding it for any kind of applications, for instance, in the agriculture area.

Despite it represents an important technological architecture for weather studies and applications, the authors can consider my recommendations below to support improving their text:

Thank you for your summary and your suggestions.  We have addressed each one below.

  1. The authors should expand their theoretical references. Currently, there are only 13 works cited, and half of them are from recent years (2020 to 2023). Please, expand and update your references.

Response: We now begin our paper with a summary of the literature that identifies the need to use integrated weather information (observations and forecasts from multiple sources) to provide warnings to reduce the risk to infrastructure and personnel. We’ve also added articles about different approaches to weather warning decision making.

  1. Did you compare your algorithmic approach with other previous existing ones? I recommend using the evidence in the literature to support it. If you produce performance metrics by testing your algorithm, you can provide it here to compare to other classification algorithms presented in the literature.

       Response: We did not find a classification algorithm similar to our problem that we could do a meaningful comparison.

  1. Please, give some details of your technological platforms, especially involving other software than ArcGIS. Which technology did you use to implement your algorithm? Which programming language, for instance?

       Response: We addressed our technological platforms in the weather classification algorithm section.

  1. Despite I understood [sic] the core contribution of the article is the classification algorithm, I missed the presentation of more related results. The results part is only focused on basic presentations of the grid in a GIS dashboard. If you have any numerical results from the algorithm application that can be presented in your text, and any other graphs, please provide them in your text with the due explanations.

       Response: We used ArcGIS to verify our algorithm correctly performed the weather warning classifications. Our partner, US-Ignite, was tasked with developing a dashboard to visualize the classifications.

  1. What are the main research contributions? You can divide them into theoretical and practical implications, for instance. Try to evidence:
    a. What does your research improve regarding the existing knowledge in classification algorithms to climatical conditions?
    b. What is the most innovative element (or elements) in your proposal, that differs from what already exists in already existing similar frameworks?
    c. What are the main contributions of your proposal for military applications?
    d. How can you apply your proposal to other areas than the military?

Response: We have inserted a new section 8 that specifically describes our research contributions.

  1. Describe the current limitations, challenges, and difficulties in the process involved in implementing and applying your framework. You can insert this description in a Final Remarks section, also incorporating the Future Work section in it.

       Response: We inserted a new section called Limitations and Challenges to address this comment.

Round 2

Reviewer 2 Report

Comments and Suggestions for Authors

The recommendations I made earlier were served in this new version, except that the authors added only a few new references based on their research topic.

Following are some references related to the topic, based on Scopus searches, which can be cited in updating the theoretical reference already built by the authors:

Smallman, H.S., Rieth, C.A. (2017). ADVICE: Decision Support for Complex Geospatial Decision Making Tasks. In: Lackey, S., Chen, J. (eds) Virtual, Augmented and Mixed Reality. VAMR 2017. Lecture Notes in Computer Science(), vol 10280. Springer, Cham. https://doi.org/10.1007/978-3-319-57987-0_37.

Stumpf, G. J., and A. E. Gerard, 2021: National Weather Service Severe Weather Warnings as Threats-in-Motion. Wea. Forecasting, 36, 627–643, https://doi.org/10.1175/WAF-D-20-0159.1.

Kaize Shi, Yusen Wang, Hao Lu, Yifan Zhu, Zhendong Niu, EKGTF: A knowledge-enhanced model for optimizing social network-based meteorological briefings, Information Processing & Management, Volume 58, Issue 4, 2021, https://doi.org/10.1016/j.ipm.2021.102564.

Michael O'Grady, David Langton, Francesca Salinari, Peter Daly, Gregory O'Hare, Service design for climate-smart agriculture, Information Processing in Agriculture, Volume 8, Issue 2, 2021, https://doi.org/10.1016/j.inpa.2020.07.003.

Chen X, Wang M, Wang S, Chen Y, Wang R, Zhao C, Hu X. Weather Radar Nowcasting for Extreme Precipitation Prediction Based on the Temporal and Spatial Generative Adversarial Network. Atmosphere. 2022; 13(8):1291. https://doi.org/10.3390/atmos13081291

Brown, B. G., L. B. Nance, C. L. Williams, K. M. Newman, J. L. Franklin, E. N. Rappaport, P. A. Kucera, and R. L. Gall, 2023: User-Responsive Diagnostic Forecast Evaluation Approaches: Application to Tropical Cyclone Predictions. Wea. Forecasting, 38, 2321–2342, https://doi.org/10.1175/WAF-D-23-0072.1.

Farzad Piadeh, Kourosh Behzadian, Albert S. Chen, Luiza C. Campos, Joseph P. Rizzuto, Zoran Kapelan, Event-based decision support algorithm for real-time flood forecasting in urban drainage systems using machine learning modelling, Environmental Modelling & Software, Volume 167, 2023, https://doi.org/10.1016/j.envsoft.2023.105772.

John Eylander, Jerry Bieszczad, Mattheus Ueckermann, Joffrey Peters, Chris Brooks, William Audette, Michael Ekegren, Geospatial Weather Affected Terrain Conditions and Hazards (GeoWATCH) description and evaluation, Environmental Modelling & Software, Volume 160, 2023, https://doi.org/10.1016/j.envsoft.2022.105606.

K. K. SINGH, KRIPAN GHOSH, S. C. BHAN, PRIYANKA SINGH, LATA VISHNOI, R. BALASUBRAMANIAN, S. D. ATTRI, SHESHAKUMAR GOROSHI, & R. SINGH. (2023). Decision support system for digitally climate informed services to farmers in India. Journal of Agrometeorology, 25(2), 205–214. https://doi.org/10.54386/jam.v25i2.2094.

I hope these references will be useful to help the authors in the mission of further improving their theoretical reference in this new round.

That done, I believe the text will be completely suitable for definitive acceptance.
